# Caring for Women Experiencing Gender-Based Violence: A Qualitative Study from the Nursing Experience

**DOI:** 10.3390/nursrep15060189

**Published:** 2025-05-28

**Authors:** Meyber González-González, Venus Medina-Maldonado

**Affiliations:** 1Gender-Based Violence Prevention Research Group (E-previo), Faculty of Health and Wellbeing, Pontifical Catholic University of Ecuador, Quito 170143, Ecuador; mjgonzalezgo@puce.edu.ec; 2Master’s Program in Care Management with a Major in Emergency Units and Intensive Care Units, Faculty of Health and Wellbeing, Pontifical Catholic University of Ecuador, Quito 170143, Ecuador; 3Emergency Service, Pediatric Obstetric Gynecological Hospital of Nueva Aurora “Luz Elena Arismendi”, Quito 170146, Ecuador; 4Center for Health Research in Latin America (CISeAL), Faculty of Health and Wellbeing, Pontifical Catholic University of Ecuador, Quito 170143, Ecuador

**Keywords:** hospital nursing services, nursing care, emergency service, gender-based violence, victimization

## Abstract

Gender-based violence is a social problem that affects the health of women in all countries, cultures, ages and economic status; its complexity requires a transdisciplinary approach. However, this study will emphasize the care provided by nursing in emergency services. **Objectives**: To explore the experiences of nursing professionals in the emergency area in relation to the approach to gender-based violence considering care management skills. **Methods**: A qualitative study with semi-structure interviews was conducted; the saturation was reached with the participation of 20 nursing professionals from emergency rooms. The study employed qualitative content analysis and the software QCAmap for organization and extraction of analysis. **Results**: The category called “Specific Approaches to Risk and Vulnerability” was characterized by its comprehensiveness, evident in its association with experiences in screening, follow-up, measures to prevent re-victimization, and ensuring privacy. The most relevant subcategories, based on the redundancy, were empathy, which encompasses affective aspects; education on forms of abuse not recognized by the victim; and multidisciplinary and multisectoral action to address patients’ needs effectively. **Conclusions**: Nursing professionals valued both the psychological and physical aspects of patients, highlighting the importance of multidisciplinary coordination and the protection of integrity. Awareness and offering help are key interventions during the activation of protocols for addressing gender-based violence.

## 1. Introduction

Gender-based violence is a public health problem that affects women in all countries, cultures, ages and economic status. The definition according to the United Nations Organization (UN) establishes that its origin is multicausal and considers any act of sexist violence that has negative consequences in the psychological, physical and sexual dimensions, affecting the autonomy of the victims in their family lives and society in general [1]. Another definition from the World Health Organization (WHO) states that gender-based violence constitutes a social problem and a violation of the human rights of those who experience it [2]. As seen in both definitions, gender-based violence refers to any abusive act that threatens or causes harm to the physical, emotional, mental, economic, sexual and reproductive health of a person or a group of people due to their sex or gender. It is important to emphasize that although girls and women suffer from gender-based violence to a greater extent, boys and men can also suffer from gender-based violence, as well as the LGBTIQ+ community [3].

According to the epidemiological perspective, the global prevalence of intimate partner violence (IPV), in 2018, 31% of women aged 15–49 who had ever been in a relationship experienced physical or sexual violence from a male partner or ex-partner at some point in their lives, while 13% experienced such violence in the past 12 months. However, these figures were further exacerbated by the pandemic. Consequently, the global agenda acknowledges insufficient progress toward achieving Goal 5 of the Sustainable Development Goals (SDGs), which aims to achieve gender equality and empower all women and girls, including eliminating all forms of violence against women and girls in both public and private spheres [4]. Women who experience gender-based violence commonly suffer from various physical health issues, including thyroid disorders, cardiovascular complications, gastrointestinal problems, diabetes, urinary tract disorders and impaired sexual health. Additionally, mental health consequences such as depression, anxiety and sleep disturbances can further increase the risk of these physical conditions due to physiological changes [5].

Particularly, Ecuador reports one of the highest rates of gender-based violence in the Latin American region, according to the results of the II Survey of Family Relations and Gender-based violence conducted by the National Institute of Statistics and Censuses (INEC): 65 out of every 100 women have suffered at least one event related to gender-based violence throughout their lives, and it also indicates that in the last twelve months, 32 out of every 100 women reported having suffered some type of violence: Physical violence 9.2%, patrimonial 6.1%, sexual 12% and psychological 25.2%. In addition, it noted that 19% of women have suffered violence in the educational sphere, 36.2% in the workplace and 20.3% in the family sphere, and the highest rate of violence against women occurs with their partner, at 42.8%. In this same survey, 48 out of every 100 women who were treated in private hospitals and clinics reported having suffered gynecological–obstetric violence [6].

On other hand, the most current and relevant research consulted in nursing on gender-based violence care had been focused on staff training; both studies pointed out the need for improve educational preparation, organizational support, skill development and interdisciplinary reflection on Intimate Partner Violence (IPV) management [7,8]. Nurses’ competencies, attitudes, beliefs and behaviors towards IPV [9] and care limitations [10,11,12]. However, to our knowledge, there is limited understanding of the ability to respond to the problem and the different actions taken in managing these cases, depending on how specific care policies and care management skills influence nursing practice.

Specifically, nurses are the first contact for women in gyneco-obstetric contexts where they are responsible for initial assessment and ongoing care. While other health professionals also have implications during the health assistance, the proximity, continuity of care and frequent interactions that nurses have with patients put them in a position to facilitate the detection of, responses to, and management of GBV cases.

From our perspective, it is important to analyze the nursing experience in a gynecology–obstetric hospital during the implementation of standards for assisting gender-based violence cases through the Purple Code Protocol. This protocol provides guidance for healthcare teams in assisting women who are experiencing situations of gender-based violence in Ecuador [13], where the emphasis is the specific approaches employed according to nurses’ reported experiences, screening, follow-up, prevention of re-victimization and the privacy aspects that guided clinical procedures.

Our study is conducted in a real clinical scenario to answer the following questions:Q1:What are the experiences of nursing professionals in the emergency area regarding the approach to gender-based violence?Q2:What are the essential nursing care management skills demonstrated during clinical practice?

The examination of the care experience allows a comprehensive analysis with the goal of strengthening nursing practice and improving the quality of care in these cases.

## 2. Materials and Methods

A qualitative study with semi-structure interviews.

### 2.1. Participant Selection

A total of 20 nursing professionals working in the emergency department participated in this study. Sampling was non-probabilistic and based on convenience. The sample size was determined by qualitative redundancy, meaning data collection continued until no new information emerged.

Prior to data collection, the study was presented during a staff meeting one month in advance. Nurses providing direct patient care, as well as the emergency department supervisor, were informed about the study. Those interested voluntarily contacted the principal investigator, who also works in the hospital’s intensive care unit.

The inclusion criteria were professional nurses with at least one year of work experience in the emergency department and signed informed consent. Exclusion criterion were nurses who declined to participate. Due to restrictions from the ethics committee, gender identity was not collected as part of this study.

### 2.2. Setting

The study was conducted in a public gynecology–obstetrics hospital located in Guamaní Parish, Quito, Ecuador. This hospital is part of the national public health network and classified as a third-level care facility. It provides specialized services across various departments, including:Emergency care with 21 beds for evaluation.Outpatient services with 21 consulting rooms.Inpatient services with 112 beds across specialties such as gynecology, high-risk obstetrics, pediatrics, maternal and neonatal intensive care units.

Interviews took place before or after the participants’ work shifts, depending on their availability.

### 2.3. Technique for Data Collection

The semi-structured interview was used and was previously designed by the main author based on the Instrument for the Multinational Study on Health and Violence against Women [14], which was validated by researchers from the Faculty of Nursing with greater experience in the subject.

The guiding questions of the semi-structured interview included the following: What is your opinion regarding the care provided to women victims of gender-based violence when the Purple Code is activated?Could you describe an experience in which you carried out or witnessed the detection of a gender-based violence case and the activation of the Purple Code?Could you explain what guidance nurses provide to female patients who are victims of gender-based violence through the Purple Code at this hospital?What does provide comprehensive nursing care mean to you in the context of Purple Code activation?In what cases do you think the Purple Code should be activated in the emergency department, and what actions should be taken when a patient refuses to accept it?

### 2.4. Data Collection

The interview procedure consisted of conducting a conversation face to face; we used a digital audio recording device. The average duration was 26.27 min, with a range of 18–40 min per interview. The selected place was the emergency services meeting room, and authorizations were previously coordinated with the service supervisor. The location was characterized by being quiet, private and uninterrupted.

After conducting the semi-structured interviews, the voice files were transcribed using the free Google Speech to Text application for review, verification of accuracy, and subsequent analysis.

### 2.5. Data Analysis

We followed the method developed by Philipp Mayring regarding qualitative content analysis, which consists of several steps: Summarizing shortened content while preserving its meaning through reduction and paraphrasing. The explanation involved detailing the content through lexicogrammatically and contextual definitions. Structuring extracted specific forms of content by defining units of analysis with coding rules and evaluating the material [15]. Specifically, we began with deductive categories derived from the theoretical framework, which in this case is the Nursing Care Management and institutional protocols for addressing gender-based violence. These predetermined categories guided our initial coding (see Table 1).

Simultaneously, in line with Mayring’s approach [15], we also allowed for the emergence of inductive categories and subcategories, which came directly from participants’ narratives in the semi-structured interviews. This dual approach allowed us to remain open to the richness of the data, particularly in capturing unexpected aspects of nurses’ experiences and meanings attributed to the implementation of the Purple Code.

We used the QCAmap software (https://www.qcamap.org/ui/en/home (accessed on 1 February 2025)) to support this process, which was designed for qualitative content analysis following Mayring’s methodology. This tool enabled us to process data flexibly and ensure transparency in category development and coding procedures.

### 2.6. Ethics Aspects

This protocol followed the principles established in the Declaration of Helsinki [16] and was approved by the Ethics Committee for Research in Human Subjects (CEISH) of the Pontifical Catholic University of Ecuador under code EO-016–2024, V2.

### 2.7. Rigor and Reflexivity

These were ensured through three key strategies implemented throughout the research process. First, during the interviews, the researchers engaged actively with participants to clarify the meaning of their responses, particularly when answers were incomplete or potentially ambiguous. This interaction allowed for real-time verification and deeper understanding of participants’ perspectives.

Second, the analysis involved collaborative categorization, triangulation, and interpretation by multiple researchers. This approach enhanced analytical rigor by reducing individual bias and ensuring that the findings reflected a shared understanding among the research team rather than the interpretation of a single researcher.

Third, the analysis was guided by two theoretical pillars: Nursing Care Management [17] and the health sector’s response to gender-based violence [13]. These theoretical frameworks informed the development of deductive categories and shaped the interpretation phase.

Rather than presuming that the response to gender-based violence was solely the result of institutional public policy, the team approached it as a reflection of nursing care management competencies. These competencies were seen as integral to professional practice and influential in guiding clinical procedures and shaping healthcare providers’ perceptions and actions.

In terms of reflexivity, the research team sought to uncover elements of professional nursing practice related to gender-based violence that are often underrepresented in the literature. This reflective stance aimed to enrich the understanding of how nurses perceive and enact care in such critical and sensitive situations.

## 3. Results

Subjects were identified using the letter “P” for PARTICIPANT, and each participant was assigned a number. Participants’ attitudes were positive, and they were always available to share their experiences during interview. From the transcripts, subordinate categories and subcategories were developed based on the standards of deductive and inductive analysis mentioned in the methodology section. The central topic called Nursing Care Management facilitated the knowledge integration in cluster groups. The figure shows the absolute count that articulates six categories and nine subcategories obtained from 20 materials where the interviews transcription were available (Figure 1).

For the purposes of this study, Nursing Care Management (NCM), which was the central topic of our analysis, “includes actions mostly related to systemic change in the health service. NCM (…) includes competencies such as network articulation and humanization to promote interprofessional action, ensuring the user’s itinerary in the network” [18]. These actions are subedited to the implementation of an institutional policy that guarantees the protection of women who have been experiencing gender-based violence but also are related to the hard and soft care manager skills that support the work of nurses in their everyday tasks.

### 3.1. Participant Characteristics

We included in this study 20 nurses (16 women and 4 men) that worked in emergency rooms with years of experience ranging from 5 to 19 years and who held the role of direct assistance to the patient in an urban public hospital.

### 3.2. Specific Approaches to Risk and Vulnerability

These approaches differ from screening, which refers to the initial, standardized detection of signs of violence. Instead, the approaches presented here reflect broader clinical reasoning, interprofessional collaboration, and patient-centered care strategies activated after or alongside screening, in response to identified or suspected violence. They involve both autonomous and interdependent nursing actions, whereby nurses draw on their professional judgment to implement immediate interventions and collaborate with other disciplines to ensure comprehensive care. These tailored actions aim to guarantee both the patient’s immediate safety and their long-term support and recovery.

**P.1:** *“For us, it means that we are going to assess the patient in all her dimensions, that is, we assess physical, psychological, and social aspects of the problems for which the patient comes to the service. Then we assess other concerns that the patient may have related with the violence situation, for example, how to file a complaint, what happens if she is financially dependent on her partner, how to manage fear and the feeling of worthlessness”*(Female).

As mentioned, the intervention encompasses a comprehensive approach that considers the psychological and physical spheres. The nurse address gender-based violence in both the private and public spheres. Then, the care targets a specific approach that responds to the needs of the patient; we found three specific approaches (Table 2).

As observed, the experiences tend to show an attitude of respect and support. They attempt to assist with recognizing the abuse, making decisions, and seeking the closest and least threatening environment for the woman who has suffered violence, with the intention of establishing a support network.

### 3.3. Screening

The following inductive category, another of the most well-founded nursing actions due to the amount of information obtained, demonstrates the experiences of nurses during the clinical role regarding the problem of gender-based violence (Table 3). The interventions highlight the defense of people’s rights, awareness of the problem, and the serious consequences for women’s health. From this set of actions emerges the inductive subcategory called *showing respect*. The experiences expressed are as follows:

The inductive subcategory *showing respect*, not forcing information, refers to a woman’s autonomy or right to free choice, but also shows the challenges faced by professionals during screening.

The analyzed testimonies revealed that the burden of implementing protective measures falls on the abused woman. In some cases, more time is needed to prepare her to act, particularly when lethal consequences are not at stake.

The inductive subcategory that appeared most in the testimonies was defined as *systematic initial assessment*, which refers to the clinical evaluation where signs and symptoms of suspected gender-based violence can be identified and is the established procedure in the hospital institution.

**P.11:** *“There are times when patients come to us for consultations simply because they are in pain, but during questioning, they never mention that they were victims of sexual, physical, or psychological abuse. However, based on the assessment, whether through observation of the patient’s care, data is collected, and we can diagnose what happened”*(Female).

However, there are still some testimonies that do not directly recognize the loss of care, but within the discursive analysis the use of the conditional “should” or “would” denotes the expression of an action that could have been performed in the past.

**P.1:** *“Cases of physical or sexual violence, and damage to the moral integrity of the person experiencing violence. A purple code should be activated in all cases”*(Female).

**P.18:** *“The ideal would be to activate the code for any instance of violence as soon as the situation is detected, but generally, physical and sexual violence trigger the purple code, and the cases are reported”*(Female).

**P.12:** *“A purple code should be activated in any case of violence, whether physical, psychological, or sexual”*(Female).

### 3.4. Follow-Up

This inductive category groups the three areas of multidisciplinary and multisectoral intervention. The derived subcategories present the professional involvement in each area. In this regard, the inductive subcategory called *Physical health* consists of direct care, which is based on procedures, treatment, and clinical care provided to the patient. The experiences expressed are as follows:

**P.13:** *“In cases of sexual violence, we apply nursing interventions such as administering prophylaxis medications against HIV and syphilis. This prophylaxis involves administering oral and intravenous medications. We provide information to the patient about why they should take the medication. Patients often refuse treatment, but we explain it to them and to their family members, who are always with them, so they can understand the importance of the treatment. We also monitor the patient until they are admitted to the institution or discharged”*(Female).

The physical approach involves taking vital signs, starting peripheral IVs, and administering the prophylaxis kit (antibiotics, retroviral, immunizations and morning-after pills). All these actions lead to the activation of the purple code, allowing for immediate, high-quality care for patients.

The subcategory called *Legal advice*, based on the experience reported by nursing professionals, consists of activating the Purple Code, informing authorities, and guiding and motivating victims on the path to initiating legal proceedings when filing a complaint. The experiences reported are as follows:

**P.4:** *“The Purple Code was activated because the girl was 14 years old and her partner was 28. All protocols were followed, including filling out the forms and calling the prosecutor’s office, just as we learned in the training (…)”*(Female).

**P.15:** *“All the Purple Code documents are activated, the police are called, and the police take the patient to the prosecutor’s office, file a report, and return”*(Female).

**P.9:** *“The detection and activation of the code is done by informing higher authorities, (…) In addition, the prosecutor’s office is notified through the hotline 911 if the patient wishes to file a complaint”*(Female).

A relevant aspect is the flexibility of the procedure. Women can initially seek services through the legal system, but there is a reverse referral to the health system.

**P.14:** *“The patient reported directly to the prosecutor’s office and was referred to our service to receive the care provided for in the Purple Code”*(Female).

Interviewees emphasize that, in legal follow-up, they activate the Purple Code, inform the Prosecutor’s Office, and support the patient throughout the reporting process. They report that they assist in completing the corresponding forms and provide follow-up to patients who decide to file a complaint. This follow-up is essential in the comprehensive management of gender-based violence cases.

In the third type of follow-up, the subcategory called *Psychosocial*, interviewees highlighted the importance of a comprehensive approach to patients, beyond the legal aspect. This perspective includes the psychological and social support provided to victims during and after the activation of the Purple Code. The experiences expressed are as follows:

**P.15:** *“We use specific cubicles to help with psychological and social support, and we are more attentive to the patient. They have equal access to healthcare, but there is priority when violence is detected”*(Female).

**P.1:** *“In the management of women, we rely on social work and psychology, we assess the patient, and we explain the risks and potential problems that could arise from violence”*(Female).

### 3.5. Avoid Re-Victimization

This inductive category reveals the importance for nursing professionals to adopt an understanding attitude when caring for women who have experienced gender-based violence, to prevent the person from re-experiencing traumatic or harmful situations during the care process. This phenomenon is called re-victimization.

It is necessary to address situations in an empathetic manner with the ability to recognize the violence or abuse and validate the person’s experience. It is necessary *not to ask the same questions repeatedly*, as this strategy leads to reliving the trauma.

Other measures implemented during the work of nursing professionals include the subcategory called *not judging or devaluing the experience*, thus avoiding making the victim feel guilty or ashamed. The experiences expressed are as follows:

**P.14:** *“Be respectful of the situation they are going through, don’t judge, and offer emotional support”*(Female).

**P.16:** * “Be more humane and put ourselves in the patient’s shoes, don’t judge (…) don’t re-victimize the patient, don’t force situations they are not willing to accept”*(Female).

**P.4:** *“Avoid asking them repeatedly what happened”*(Female).

**P.2:** *“(…) Re-victimization occurs in patients when we ask repeatedly what happened, which is why it is important to handle ourselves as well as the entire team”*(Female).

The interviewees’ experiences stand out for their *empathy*, which is the basis for creating an environment of trust and providing subsequent emotional support. Additionally, the last testimony demonstrates a key element in preventing the re-victimization of women experiencing gender-based violence in an emergency department, which would be substantially reduced if there is good communication and trust among the team members who approach the patient.

### 3.6. Privacy

The Privacy category addresses ethical aspects derived in the subcategory respect for confidentiality, which relates to the importance of keeping the victim’s information confidential and protected and looking for a private space in the service, which implies providing the victim with a private and quiet space. The participants’ stories are presented below:

**P.5:** *“At this hospital, at least their integrity and confidentiality have been respected (…). They are treated anonymously in this situation”*(Female).

**P.3:** *“We try to keep this information secure and private for the patient”*(Female).

However, some experiences indicate that *infrastructure limitation* can sometimes be a problem in providing adequate care.

**P.8:** *“Sometimes we don’t have a specific space to care for these patients, (…) so we move them to an unoccupied cubicle in the observation room”*(Female).

**P.15:** *“We should have more cubicles because, at times, patients are placed in three adjacent beds, allowing those nearby to overhear their conversations. This prevents them from fully sharing their experiences. We need more private and discreet spaces (…)”*(Female).

These feedback loops are essential as they provide insights into the infrastructure needed to create a safe and respectful environment. They ensure that women feel comfortable and confident sharing their experiences without fear of judgment or exposure.

### 3.7. Effectiveness of the Protocol

This deductive category addresses nursing professionals’ perceptions of the policy for protecting women who are experiencing violence. It highlights the importance of *essential care*, which involves coordinated actions to comply with this regulation, aimed at providing care, protection and recovery for victims (Table 4). Nurses must be proficient in managing documentation and ensuring complete and accurate records for legal interventions following the activation of the Purple Code. The subcategories derived from the analysis were *essential care, documentation and nursing record-keeping*, as well as the *activation and implementation of Ministry of Health regulations*. The experiences reflect the priority clinical actions and procedures applied in practice.

The participants’ perceptions indicate awareness of the mandatory regulations, understanding of their role in activating the Purple Code, and the significance of delivering optimal and comprehensive care to address this public health issue, which is evident daily in the emergency department.

## 4. Discussion

The relevant finding of this research is the specific approaches that result from the autonomous action of the nurse, where the protection of the integrity of women experiencing gender-based violence and the ability to be alert to signs of abuse not expressed by patients (psychological or physical) from the first contact are prioritized. Collaborative action is seen in the understanding that the complexity of the problem demands coordination with the health team and other sectors of society. This is in line with a previous study in which it was shown that the most committed careers in the health sector for the assistance of this problem are nurses, doctors and psychologists, since they have a fundamental role in the care and management of victims of gender-based violence [19].

The aforementioned point is closed related to communication skills, a key soft skill in nursing care management. This was evident in the nurses’ narratives during the interviews. Three elements were implicit in the communication with women experiencing gender-based violence: active listening, watching body language, and providing information. At the organizational level, communication involved multidisciplinary coordination and close collaboration with the healthcare team. According to the literature, this type of care not only helps address immediate health needs but also promotes the empowerment of victims through self-advocacy and multidisciplinary support [20,21,22].

Other soft skills that emerged during analysis were the affective and emotional component during the care. In this research, nurses demonstrated expressions of sensitivity and empathy, i.e., the ability to share the feelings of others. The experience applied from gender perspective consisted of not judging, not devaluing the experience, and not re-victimizing. This effect is achieved in nursing professionals when they examine and question their own beliefs about gender and gender-based violence. According to a previous study, nurses recognize that their personal ideology conditions both the response to gender-based violence and their attitudes towards patients [9,23]. Furthermore, institutional policy with uninterrupted training and an educational approach that includes personal reflection must be implemented.

In this study, nurses demonstrated that their autonomy in actions derived from specific approaches. These included screening, education, follow-up, essential therapeutic care and care management tailored to patient needs. These are considered as hard skills in nursing care management because they correspond to expertise and knowledge. The stories analyzed show decision-making based on critical analysis and knowledge acquired with the protocol implemented in the service. A research has highlighted the significance of specific training in identifying the signs and symptoms of abuse. A solid foundation of specialized knowledge is crucial for delivering effective care to victims of gender-based violence [24].

In addition, this study also examined ethical and safety issues in care provision. Specifically, in privacy management, some actions were outlined, including the commitment to confidentiality and protection of information, maintaining confidentiality and managing consent in the event of a patient’s refusal to receive services. This is why our findings align with a prior study which concluded that the health team in the emergency service received specific training in gender-based violence and demonstrated in-depth knowledge of the signs and symptoms of abuse [20,21,22]. Additionally, both in the published study and in the perceived experience of the participants of this research, it is noted that the institution is carrying out continuous training [23].

One barrier identified in this study to providing care is infrastructure, and this becomes a challenge to ensuring privacy, as expressed by several participants. On this subject, a previous study indicates that the barriers include a lack of training, education, time, privacy, guidelines, policies and employer support, with training and education being the most frequent [25]. However, one difference found in our study was the stories in which nurses expressed the efforts to search for a private place in the emergency room. Our study confirmed that the nurses keep a positive attitude toward protecting women who are survivors of gender-based violence. This includes an active commitment to detecting cases and activating the “Purple Code” protocol, which is part of a recent public policy implemented within the health institution.

The second relevant finding was related to the usefulness of the protocol. The interviewed professionals expressed the importance of applying policies and laws in relation to gender-based violence, and this is in line with the findings of Öhman et al., 2020 [26], who indicated that the health sector has a legal obligation to apply and provide adequate medical care to victims of violence within close relationships [26]. The effective implementation of these policies not only improves the quality of care but also contributes to the prevention and reduction of gender-based violence, reaffirming its importance as a public health problem that requires a comprehensive and multidisciplinary response.

Subsequently, it is clear that fundamental nursing skills for the assistance of women experiencing gender-based violence require a combination of both hard skills and soft skills [7,8,10,17,19]. Also, the findings showed complementary treatment of the problem with a gender perspective, training for the healthcare team, and a clear institutional policy to ensure comprehensive support. The challenge in providing this care lies in the need for professionals to show genuine interest in the victims’ stories and their health conditions, and to enhance women’s awareness, especially when they deny experiencing violence.

### 4.1. Limitation

A limitation of this research is that the study focused on the experience of a single healthcare institution’s emergency room, which prevents generalization. However, this study, based on the experience reported by nurses, shows that guiding case management, providing ongoing training, and defining pathways for hospital-based treatment of gender-based violence can overcome barriers to healthcare. The hospital selected for the participation of nursing professionals stands out because the population served consists of patients with gynecological and obstetric complications, and the way professionals and institutions classify and monitor patients also influences the findings.

### 4.2. Implication for Clinical Practice

Currently, gender-based violence is considered a social and public health problem, especially in Ecuador, the country where this research was carried out. The total number of femicides in 2024 was around 271 cases [27]. In emergency rooms, there is a considerable advantage for the management of care for women experiencing gender-based violence because the place is an opportunity for detection and follow-up. The participation of nursing professionals is highlighted since most can adequately identify cases of gender-based violence, are clear about the routes to follow, and support the existence of the institutional protocol.

## 5. Conclusions

The health institution analyzed demonstrated the implementation of a protocol that follows the public policy of providing protection to patients identified as experiencing gender-based violence. The service’s outstanding characteristic was its comprehensiveness and multidisciplinary approach. The nurses and their experiences demonstrated not only sensitivity to the issue but also specialized knowledge, documentation skills, appropriate care decisions and an understanding of legal guidelines.

The accounts obtained from the nurses showed a collective awareness of the problem and a commitment to reporting cases of gender-based violence. Nevertheless, the nurses’ decision to report a case depends on the risk assessment provided and the level of danger faced by the victim. The response varied in cases involving minors or due to the lethality of the violence.

Hospital emergency rooms are an ideal place to detect the problem and refer patients to primary or complementary services. In a gynecological–obstetric hospital where the patients are entirely women, a protocol for treating this type of patient is expected to be implemented. However, the soft skills of care management were also clear in clinical practice, achieving human connection with patients, their families, and coordinating work with the healthcare team.

## Figures and Tables

**Figure 1 nursrep-15-00189-f001:**
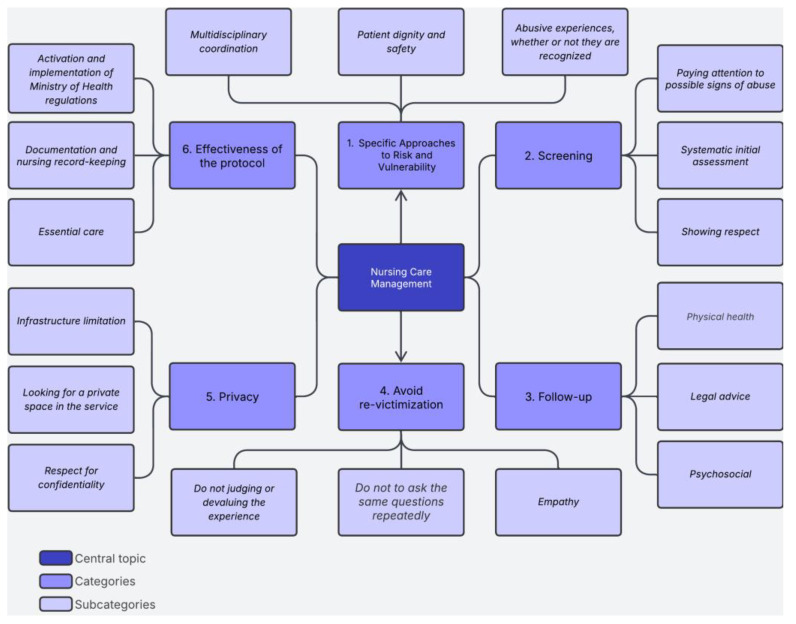
Knowledge integration cluster. **Source:** Prepared by the author after the interview categorization process.

**Table 1 nursrep-15-00189-t001:** Deductive categories determined.

Categories	Definitions	Anchor Expressions
C1: Specific Approaches to Risk and Vulnerability	This category aimed at addressing the complex needs and vulnerabilities of women in emergency contexts due to gender-based violence.	Comprehensive care is provided, the victim’s integrity is guaranteed, and work is done in conjunction with other professionals and services.
C2: Effectiveness of the Protocol	It encompasses the opinion of the relevant procedures carried out to provide comprehensive care to the victim of gender-based violence.	The multidisciplinary approach is carried out, notification is submitted, and the protocol is activated.

Source: Triangulation of the researchers based on theoretical discussion and expected results.

**Table 2 nursrep-15-00189-t002:** Specific nursing approaches to risk and vulnerability in gender-based violence: subcategories and illustrative quotations.

Subcategories	Quotations
*Multidisciplinary coordination* shows the interdependent actions in nursing practice that ensure more efficient care and a more supportive environment.	**P.2:** *“We are a team of nurses, physicians, psychologists, and social workers, as we all need to interact to ensure the patient has a better outcome when the protocol is activated” (Female).* **P.3:** *“Care is based on addressing the patient’s needs, and we participate as member of the health team, the team provides medical assessment, psychological support, nursing care, and social work assessment for both the victim and their families” (Female).*
*Patient dignity and safety* expresses the actions that keep people safe from potential or real harm while they remain in the emergency room. This action is categorized as an independent intervention.	**P.4:** *“If we identify a person who has been abused, the first thing we do is maintain the patient’s integrity. That is, we isolate the patient. Even if there are only a few of them, we look for an individual cubicle. If the patient is accompanied by the abuser, we separate them from that person. There, we activate the purple code, perform all the laboratory tests, administer prophylactic medication, and call the Prosecutor’s Office to facilitate the patient’s complaint” (Woman).*
*Abusive experiences, whether or not they are recognized* displays the skills and abilities demonstrated by nursing staff when they are aware of abuse or when it is suspected. Specifically, the independent nursing interventions during the activation of the purple code include raising awareness and offering help to the patient through education.	**P.5:** *“She received initial help or initial care, where she felt calm and could share with me her sadness (…) she cried a lot and tried to talk as much as she could. The relevant services were contacted and coordinated so she could move forward and leave the abusive relationship” (Female).* **P.6:** *“When the patient refuses (…) she is informed of what this could lead to (…) , what problems will arise afterward if she does not decide to file a report” (Male).* **P.7:** *“If there is a suspicion, as I told you, we try to investigate privately without a companion and we warn to the treating physician, so that, in the in-depth assessment, they can find out more” (Male).* **P.8:** *“If a patient were to refuse to participate in the activation of the purple code, it is already established in that part; it is not true that she is under 14 years of age. Obviously, an approach would have to be made, verify if he is with a family member who is supportive and, in that case, talk to a trusted family member who has come” (Female).*

Source: Interview transcriptions.

**Table 3 nursrep-15-00189-t003:** Screening: subcategory and illustrative quotations.

Subcategories	Quotations
*Showing respect*	**P.7:** *“Quite complicated. For me, the most difficult thing is trying to uncover or uncover what the patient is suffering from” (Male).* **P.9:** *“When the patient or victim refuses to accept it, we still have to manage the entire protocol and (…) the respective report is made (…) they are asked to sign and certify their refusal of this type of care” (Male).* **P.10:** *“If the patient doesn’t want it, her consent would be respected, because that was the case. We had the Purple Code activated, and she requested voluntary discharge and left. We only had her sign the consent and voluntary discharge because we can’t go against the patient’s will” (Female).*

Source: Interview transcriptions.

**Table 4 nursrep-15-00189-t004:** Effectiveness of the protocol: subcategories and illustrative quotations.

Subcategories	Quotations
*Essential care*	**P.17:** *“educate the patient and their family equally about the steps they need to follow, including consultations with social work, psychology, and other necessary services, as well as ensuring follow-up after the complaint and care”* (Female).
*Documentation and nursing record-keeping*	**P.10:** *“We follow up by submitting the notification using Form 094, which specifies the legal regulations applicable to this type of patient”* (Female).
*Activation and implementation of Ministry of Health regulations*	**P.9:** *“The detection of cases and activation of the code are carried out by informing the higher au-thorities (…). Additionally, the prosecutor’s office is notified through the ECU 911 emergency line if the patient wishes to file a complaint”* (Female). **P.13:***“Now, with the multidisciplinary approach of professionals toward the patient-user, the re-maining workflows will be determined while adhering to our protocol, as mandatory reporting is required”* (Female).

Source: Interview transcriptions.

## Data Availability

Data are unavailable due to privacy of participants and ethical restrictions.

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
