# Peer review of "Caring for Women Experiencing Gender-Based Violence: A Qualitative Study from the Nursing Experience"

_nursrep, 2025, doi:10.3390/nursrep15060189_

Round 1

Reviewer 1 Report

Comments and Suggestions for Authors

General: Please use gender-based violence rather than gender violence
Abstract: Gender violence is not a health problem…it is the social and criminal cause of numerous severe health problems. While perhaps this seems subtle, it frames everything we know and do about it. The authors know this, as evidenced by their review of the literature. It is just written wrong in the abstract.
Line 21: “Results: The category 'Approach to Gender Violence' demonstrated the strongest theoretical foundation and density.” If the authors had used content analysis, they would not have addressed “theoretical foundations.” In addition, “approach” isn’t a theme, but a categorization of the data, probably related to the interview.
Line 80: purple code protocol needs explanation and a reference
Line 91-92: “epistemological and methodological current to understand the meaning and experience”: something is missing or incorrect here
Line 97: “theoretical saturation” references grounded theory, and this analytic design was phenomenological. It was analytic saturation, or perhaps more accurately, qualitative redundancy that was reached. Moreover, content coding is inappropriate for phenomenology, which aims to find deep and rich experiential meaning. I see that the authors closely followed the Qualitative content analysis approach outlined in the book by Philipp Mayring, who asserts that you can use content analysis, hermeneutics, and phenomenological approaches using content analysis. I don’t agree, but since it was the entire framework of this study, it would mean that the paper wasn’t publishable, however, they followed this method to the letter, and they did nothing wrong. Perhaps a way to save the paper is to address this issue in the discussion. The issue is whether one can find and explore meaning and report it through counting them as numbers. 
Lines 100-105: We need more details about the questions in the interview
Line 52: The authors reference the global agenda and Goal 5, but don’t explain it
Line 173: The authors refer to deductive categories. Is this project starting with a theory? Then on line 193, the authors refer to an inductive category…Im lost. What is the theory, and is it that they are using inductive analysis to flesh out the theory?  How is this phenomenological? If it is the nursing process that is the “theory” that they used as their deductive general categories, then, follow-up, on line 270, would be a deductive theoretical concept, right? Also, on line 357, that effectiveness section is “evaluation” as a deductive theoretical concept? If the nursing process is somehow guiding the analysis, it just needs to be explained in the analysis section. It isn’t a problem, and is consistent with the book they used to guide their analysis, but all of that needs to be broken down for us.

Finally, while they are calling this content analysis (the phrase Mayring uses), this reads like thematic analysis, and they use the word theme throughout. I suggest that they did use thematic analysis. There is ample richness in the findings to support this assertion. In that case, I think they were guided by Philipp Mayring approach but adapted it to thematic analysis to preserve the richness and meaning in the analysis.

Author Response

3. Point-by-point response to Comments and Suggestions for Authors

Comments 1: General: Please use gender-based violence rather than gender violence.

Response 1: Thank you for your feedback. We agree with this comment and have made the suggested change throughout the entire document. 

Comments 2: Abstract: Gender violence is not a health problem…it is the social and criminal cause of numerous severe health problems. While perhaps this seems subtle, it frames everything we know and do about it. The authors know this, as evidenced by their review of the literature. It is just written wrong in the abstract.

Response 2: Agree. We have modified. Line 13.

Comments 3: Line 21: “Results: The category 'Approach to Gender Violence' demonstrated the strongest theoretical foundation and density.” If the authors had used content analysis, they would not have addressed “theoretical foundations.” In addition, “approach” isn’t a theme, but a categorization of the data, probably related to the interview.

Response 3: We have addressed this point accordingly and we have changed the term for clarification of the category. 'Specific Approaches to Risk and Vulnerability' was conceived as a category, and this has been clarified in the revised manuscript. Resume, line 164, line 227.

Comments 4: Line 80: purple code protocol needs explanation and a reference.

Response 4: We have addressed this point accordingly. “From our perspective, it is important to analyze the nursing experience in a gynecology-obstetric hospital during the implementation of standards for assisting gender-based violence cases through the Purple Code Protocol. This protocol provides guidance for healthcare teams in assisting women who are experiencing situations of gender-based violence in Ecuador [13]” Lines 80-84 

Comments 5: Line 91-92: “epistemological and methodological current to understand the meaning and experience”: something is missing or incorrect here”

Response 5: Agree. We agree with the reviewer’s observation. To avoid confusion and ensure consistency with terminology commonly used in nursing science, we have removed the original phrase. Additionally, based on how similar studies using this method are presented in the literature, we revised the description of the study design to read: “A qualitative study based on semi-structured interviews.” This change helps reduce the methodological tension and aligns with accepted reporting practices in qualitative nursing research. The revision was made in Line 101.  

Comments 6: Line 97: “theoretical saturation” references grounded theory, and this analytic design was phenomenological. It was analytic saturation, or perhaps more accurately, qualitative redundancy that was reached. Moreover, content coding is inappropriate for phenomenology, which aims to find deep and rich experiential meaning. I see that the authors closely followed the Qualitative content analysis approach outlined in the book by Philipp Mayring, who asserts that you can use content analysis, hermeneutics, and phenomenological approaches using content analysis. I don’t agree, but since it was the entire framework of this study, it would mean that the paper wasn’t publishable, however, they followed this method to the letter, and they did nothing wrong. Perhaps a way to save the paper is to address this issue in the discussion. The issue is whether one can find and explore meaning and report it through counting them as numbers.

Response 6: Thank you for this observation. We agree with your comment and have replaced the term “theoretical saturation” with “qualitative redundancy” to improve our methodological approach. Line 105. 

Comments 7: Lines 100-105: We need more details about the questions in the interview

Response 7: Thank you for your valuable comment. We have included a brief description of the semi-structured interview questions and it has been incorporated into the Methods section. “The guiding questions of the semi-structured interview included: What is your opinion regarding the care provided to women victims of gender-based violence when the Purple Code is activated? Could you describe an experience in which you carried out or witnessed the detection of a gender-based violence case and the activation of the Purple Code? Could you explain what guidance nurses provide to female patients who are victims of gender-based violence through the Purple Code at this hospital? What does providing comprehensive nursing care mean to you in the context of Purple Code activation? In what cases do you think the Purple Code should be activated in the emergency department, and what actions should be taken when a patient refuses to accept it?” lines 131-139.  

Comments 8: Line 52: The authors reference the global agenda and Goal 5, but don’t explain it

Response 8: Agree. We have revised the sentence to provide a clearer explanation of Goal 5. The revised text now reads: “Consequently, the global agenda acknowledges insufficient progress toward achieving Goal 5 of the Sustainable Development Goals (SDGs), which aims to achieve gender equality and empower all women and girls, including eliminating all forms of violence against women and girls in both public and private spheres [4].” Line 50. 

Comments 9: Line 173: The authors refer to deductive categories. Is this project starting with a theory? Then on line 193, the authors refer to an inductive category…Im lost. What is the theory, and is it that they are using inductive analysis to flesh out the theory?  How is this phenomenological? If it is the nursing process that is the “theory” that they used as their deductive general categories, then, follow-up, on line 270, would be a deductive theoretical concept, right? Also, on line 357, that effectiveness section is “evaluation” as a deductive theoretical concept? If the nursing process is somehow guiding the analysis, it just needs to be explained in the analysis section. It isn’t a problem, and is consistent with the book they used to guide their analysis, but all of that needs to be broken down for us.

Response 9: We greatly appreciate this comment and the opportunity to clarify our analytical approach. In response, we have expanded the Analysis section to explain our use of Philipp Mayring’s qualitative content analysis, which integrates both deductive and inductive procedures in a systematic way. “Specifically, we began with deductive categories derived from the theoretical framework, which in this case is the Nursing Care Management and institutional protocols for ad-dressing gender-based violence. These predetermined categories guided our initial coding (see Table 1).

Simultaneously, in line with Mayring’s approach, we also allowed for the emergence of inductive categories and subcategories, which came directly from participants' narratives in the semi-structured interviews. This dual approach allowed us to remain open to the richness of the data, particularly in capturing unexpected or nuanced aspects of nurses’ experiences and meanings attributed to the implementation of the Purple Code. We used the QCAmap software to support this process, which was designed for qualitative content analysis following Mayring’s methodology. This tool enabled us to process data flexibly and ensure transparency in category development and coding procedures.” in Lines 160-173.

Explanation: Our manuscript at the beginning was informed by the phenomenological interest in understanding lived experiences, and it is important to explain that the qualitative content analysis as per Maying does not conflict with phenomenological aims, especially when exploring the meaning structures of experiences from the participant’s perspective. However, we have updated the Analysis section accordingly to improve the integration of theoretical and data-driven category development as well as the design of the study.

Additionally, we have revised the terminology used in the manuscript to maintain conceptual clarity and consistency. We replaced terms such as ´nursing process´ and ´nursing performance´ with Nursing Care Management, which more accurately reflects the theoretical framework that guided our analysis. This change helps to avoid deviations and ensures that the deductive categories align with the central aims of this study. Lines 161, 188, 193, 207, and 214.

Comments 10: Finally, while they are calling this content analysis (the phrase Mayring uses), this reads like thematic analysis, and they use the word theme throughout. I suggest that they did use thematic analysis. There is ample richness in the findings to support this assertion. In that case, I think they were guided by Philipp Mayring approach but adapted it to thematic analysis to preserve the richness and meaning in the analysis.

Response 10: Thank you for this useful and constructive comment. We agree that our findings contain a level of richness consistent with thematic analysis. In response, we have clarified the methodological explanation in the Analysis section. We continue to frame our work within Philipp Mayring’s qualitative content analysis, which offers a structured yet flexible framework, allowing for both deductive and inductive category development. However, we recognize that elements of thematic analysis such as the focus on meaning, theme development, and interpretive depth were also present in our approach. To ensure methodological consistency and transparency, we revised the Analysis section to better reflect this integration. Specifically, we emphasized how the deductive framework was informed by theoretical assumptions (nursing care management and health sector responses to gender-based violence), while still allowing inductive insights to emerge from the participants’ narratives. We also removed terminology that might be associated with other analytic methods to avoid confusion. Lines 160-173. 

4. Response to Comments on the Quality of English Language

Point 1:

Response 1:  This manuscript will be submitted to MDPI’s language editing services to improve the quality of the English language. We have used language support tools during the writing process, and should the manuscript be accepted, we are committed to ensuring the final version meets the journal’s standards.

Reviewer 2 Report

Comments and Suggestions for Authors

Thank you so much for this interesting and timely study. I really enjoyed reading it. I think this paper is well written, but there are a few areas that need to be strengthened. I have a few comments that I have discussed below.

Abstract: I think the Results section of the abstract needs to be reworked a bit to include high-level results. I feel the way it is written at the moment can be a bit confusing. For example, the sentence "The category 'Approach to Gender Violence' demonstrated the strongest theoretical foundation and density." As a reader, I would be a bit confused by the "Approach to Gender Violence" category. I feel an abstract on its own should communicate what the study is about in a nutshell, in a way that encourages the reader to read more. I hope the authors will take this into account when revising the paper.

Introduction: I enjoyed reading the introduction. The background information is very detailed and shows that a lot of extensive literature review has been conducted. However, I feel like the rationale of this study can be strengthened. You have done some good work on that, which is commendable, but I feel you can flesh out the section a bit to show why it is particularly important to focus on nursing professionals in this instance, when there are other health professionals. Is it because they are the first point of contact for most survivors? I hope this makes sense.

Methods: I have a few comments on this section. I think there is a need for more information on the inclusion criteria. Were you targeting nurses who self-identify as females or all genders? I might have missed it, but I was not sure if you included nurses working in private or public hospitals or both. You mentioned the hospital, but is it a private or public hospital? As someone not familiar with the study setting, I am curious to know. I also think the Methods section can benefit from reorganizing ideas, perhaps with some subheadings to make things clearer. I will leave this to the authors to decide.

Results: I think it will be good to also include the demographics of the participants included in the study, e.g., gender, years of experience. Quite a detailed Results section, which I enjoyed reading. I also think this section can benefit from some reorganizing of points, particularly the 'Approach' theme. As I was reading through, I felt like after assessment, maybe we talk about paying attention to possible signs, followed by abuse experience whether or not they are recognizable, then patient dignity and safety, etc. I felt this gives a good flow of ideas in this category. I am also curious to know if survivors disclosed violence at the first go or if they come with different issues, e.g., injuries, and then nurses 'identify' GBV/IPV during assessment, especially when accompanied by the abuser.

Discussion: I think this section can benefit from some subheadings, e.g., implications and limitations, for easier flow of ideas.

Copyediting: Lastly, there are a few grammatical errors in the paper that might require some copyediting.

All the best with the paper!!!!

Author Response

3. Point-by-point response to Comments and Suggestions for Authors

Comments 1: Abstract: I think the Results section of the abstract needs to be reworked a bit to include high-level results. I feel the way it is written at the moment can be a bit confusing. For example, the sentence "The category 'Approach to Gender Violence' demonstrated the strongest theoretical foundation and density." As a reader, I would be a bit confused by the "Approach to Gender Violence" category. I feel an abstract on its own should communicate what the study is about in a nutshell, in a way that encourages the reader to read more. I hope the authors will take this into account when revising the paper.

Response 1: Thank you for your feedback. We agree with your comment and have revised the abstract for clarity and accessibility. The revised text now reads:

“Results: The category 'Specific Approaches to Risk and Vulnerability’ was characterized by its comprehensiveness, reflected in participants’ experiences related to screening, follow-up, prevention of re-victimization, and ensuring privacy. The most relevant subcategories, based on redundancy, were empathy (encompassing affective aspects), education about forms of abuse not recognized by victims, and the importance of multidisciplinary and multisectoral collaboration in addressing patients’ needs.” Lines 20-25.

Comments 2: Introduction: I enjoyed reading the introduction. The background information is very detailed and shows that a lot of extensive literature review has been conducted. However, I feel like the rationale of this study can be strengthened. You have done some good work on that, which is commendable, but I feel you can flesh out the section a bit to show why it is particularly important to focus on nursing professionals in this instance, when there are other health professionals. Is it because they are the first point of contact for most survivors? I hope this makes sense.

Response 2: Thank you for your constructive feedback. Specifically, nurses are the first contact for women in gyneco-obstetric contexts where they are responsible for initial assessment and ongoing care. While other health professionals have also implications during the health assistance, the proximity, con-tinuity of care, and frequent interactions that nurses have with patients is a position to facility the detection, responses, and management of GBV cases” Lines 80-84.

Comments 3: Methods: I have a few comments on this section. I think there is a need for more information on the inclusion criteria. Were you targeting nurses who self-identify as females or all genders? I might have missed it, but I was not sure if you included nurses working in private or public hospitals or both. You mentioned the hospital, but is it a private or public hospital? As someone not familiar with the study setting, I am curious to know. I also think the Methods section can benefit from reorganizing ideas, perhaps with some subheadings to make things clearer. I will leave this to the authors to decide.

Response 3: Thank you for your helpful suggestions. We have reorganized the Methods section using subheadings and included additional information regarding the setting and inclusion criteria. The disaggregation information by sex was specified in methods. However, gender identity data were not collected, as this variable was not approved by the ethics committee. We also clarify that the study was conducted in a public hospital. Lines 101-173.

Comments 4: Results: I think it will be good to also include the demographics of the participants included in the study, e.g., gender, years of experience. Quite a detailed Results section, which I enjoyed reading. I also think this section can benefit from some reorganizing of points, particularly the 'Approach' theme. As I was reading through, I felt like after assessment, maybe we talk about paying attention to possible signs, followed by abuse experience whether or not they are recognizable, then patient dignity and safety, etc. I felt this gives a good flow of ideas in this category. I am also curious to know if survivors disclosed violence at the first go or if they come with different issues, e.g., injuries, and then nurses 'identify' GBV/IPV during assessment, especially when accompanied by the abuser.

Response 4: Thank you for your constructive feedback. We have added the demographic information of the nursing staff interviewed to improve our findings. In addition, we have renamed the category Approach to Specific Approaches to Risk and Vulnerability in order to clearly distinguish it from initial screening procedures and to highlight the autonomous and interdependent aspects of nursing interventions. Lines 223-248.

“3.1 Participants characteristics

We included in this study 20 nurses (16 women and 4 men) that worked in emergency room with years of experiences ranging from 5 to 19 years and held the role of direct assistance to the patient in an urban public hospital.

3.1. Specific Approaches to Risk and Vulnerability

These approaches differ from screening, which refers to the initial, standardized detection of signs of violence. Instead, the approaches presented here reflect broader clinical reasoning, interprofessional collaboration, and patient-centered care strategies activated after or alongside screening, in response to identified or suspected violence. They involve both autonomous and interdependent nursing actions, whereby nurses draw on their professional judgment to implement immediate interventions and col-laborate with other disciplines to ensure comprehensive care. These tailored actions aim to guarantee both the patient’s immediate safety and their long-term support and recovery”.

Comments 5: Discussion: I think this section can benefit from some subheadings, e.g., implications and limitations, for easier flow of ideas.

Response 5: Thank you for your suggestion. We agree with your observation and have added subheadings such as Implications and Limitations in the Discussion section. Lines 465 y 474.

4. Response to Comments on the Quality of English Language

Point 1: Copyediting: Lastly, there are a few grammatical errors in the paper that might require some copyediting.

Response 1: Thank you for your comment. This manuscript will be submitted to MDPI’s language editing services to improve the quality of the English language. We have used language support tools during the writing process, and should the manuscript be accepted, we are committed to ensuring the final version meets the journal’s standards.

Reviewer 3 Report

Comments and Suggestions for Authors

Dear Authors,
             Thank you for your valuable contribution to the field of nursing and gender-based violence response. Your qualitative study presents a highly relevant and original exploration of nurses' experiences in a real emergency clinical setting.
Strengths of the study: The focus on nursing care management in the context of gender-based violence is both timely and underexplored in current literature. The integration of hard and soft skills into the analysis provides depth and applicability to practice. Use of the Purple Code protocol and its interpretation is a valuable model for other healthcare institutions.
Points for improvement: The presentation of the results, while rich in content, could benefit from greater conciseness and improved organization to enhance clarity. Consider adding visual aids (e.g., thematic maps or tables) to better illustrate the relationships among subcategories. The discussion could further elaborate on transferability to other contexts beyond the single institution.
Overall, the study is a commendable and important effort that deepens the understanding of nursing roles in violence prevention and care. With some refinement, it can have even greater impact.
Kind regards

Author Response

3. Point-by-point response to Comments and Suggestions for Authors

Comments 1: The presentation of the results, while rich in content, could benefit from greater conciseness and improved organization to enhance clarity. Consider adding visual aids (e.g., thematic maps or tables) to better illustrate the relationships among subcategories.

Response 1: Thank you for your constructive feedback. Some findings are concise enough to be presented in a table, while others are significant enough to be highlighted in the main text as a central part of the results section. Lines 241-393.

Comments 2: The discussion could further elaborate on transferability to other contexts beyond the single institution.

Response 2: Thank you for this useful and constructive comment. ”As we reflect on the scalability of institutional strategies, the nursing activities evaluated after the implementation of the Purple Protocol may serve as a model for other hospitals or health centers, particularly in Latin America or in countries with similar public health frameworks. Combining legal compliance with soft skills development and intersectoral coordination is adaptable and could enhance victim support in various healthcare settings.” Lines 468-473.

Response to Comments on the Quality of English Language

Point 1: Copyediting: Lastly, there are a few grammatical errors in the paper that might require some copyediting.

Response 1: This manuscript will be submitted to MDPI’s language editing services to improve the quality of the English language. We have used language support tools during the writing process, and should the manuscript be accepted, we are committed to ensuring the final version meets the journal’s standards.
